# Altered Retrograde Signaling Patterns in Breast Cancer Cells Cybrids with H and J Mitochondrial DNA Haplogroups

**DOI:** 10.3390/ijms23126687

**Published:** 2022-06-15

**Authors:** Steven Chang, Lata Singh, Kunal Thaker, Sina Abedi, Mithalesh K. Singh, Tej H. Patel, Marilyn Chwa, Shari R. Atilano, Nitin Udar, Daniela Bota, Maria Cristina Kenney

**Affiliations:** 1Gavin Herbert Eye Institute, University of California Irvine, Irvine, CA 92697, USA; sychang2@uci.edu (S.C.); drlata.singh@aiims.edu (L.S.); kthaker@uci.edu (K.T.); sabedi1@uci.edu (S.A.); mithales@hs.uci.edu (M.K.S.); tej480@gmail.com (T.H.P.); mchwa@hs.uci.edu (M.C.); satilano@hs.uci.edu (S.R.A.); greatbioinformatics@yahoo.com (N.U.); 2Department of Neurology, Neuro-Oncology Division, University of California Irvine, Irvine, CA 92697, USA; dbota@hs.uci.edu; 3Department of Pathology and Laboratory Medicine, University of California Irvine, Irvine, CA 92697, USA

**Keywords:** cybrids, mitochondria, cGAS-STING pathway

## Abstract

The aim of this study was to determine the role of retrograde signaling (mitochondria to nucleus) in MCF7 breast cancer cells. Therefore, in the present study, MCF7-H and MCF7-J cybrids were produced using the mitochondria from the same H and J individuals that were already used in our non-diseased retinal pigment epithelium (ARPE19) cybrids. MCF7 cybrids were treated with cisplatin and analyzed for cell viability, mitochondrial membrane potential, ROS, and expression levels of genes associated with the cGAS-STING and cancer-related pathways. Results showed that unlike the ARPE19-H and ARPE19-J cybrids, the untreated MCF7-H and MCF7-J cybrids had similar levels of ATP, lactate, and OCR: ECAR ratios. After cisplatin treatment, MCF7-H and MCF7-J cybrids showed similar (**a**) decreases in cell viability and ROS levels; (**b**) upregulation of *ABCC1*, *BRCA1* and *CDKN1A/P21*; and (**c**) downregulation of *EGFR*. Cisplatin-treated ARPE19-H and ARPE19-J cybrids showed increased expression of six cGAS-STING pathway genes, while two were increased for MCF7-J cybrids. In summary, the ARPE19-H and ARPE19-J cybrids behave differentially from each other with or without cisplatin. In contrast, the MCF7-H and MCF7-J cybrids had identical metabolic/bioenergetic profiles and cisplatin responses. Our findings suggest that cancer cell nuclei might have a diminished ability to respond to the modulating signaling of the mtDNA that occurs via the cGAS-STING pathway.

## 1. Introduction

The mitochondria (mt) are unique organelles in that they possess their own DNA. The mtDNA is maternally inherited and can be classified into haplogroups which represent ancestral origins of populations. The haplogroups are defined by an accumulation of specific single nucleotide polymorphisms (SNPs) that represent a geographic origin for a population. The most common European haplogroup is H, representing over 40% of the population [1]. The J haplogroup, representing populations that originated in Northern Europe, account for approximately 9% of the population [1]. The mitochondria are critical for oxidative phosphorylation (OXPHOS) and bioenergetics of all cells.

Belizzi et al. compared oxygen consumption rates of test subjects with different European haplogroups and reported that the H haplogroup subjects had higher O_2_ consumption rates while the J haplogroup rates were lower, suggesting higher glycolysis level in the latter group [2]. Similarly, Larsen et al. found that test subjects with the H haplogroup mtDNA had increased oxygen consumption rates compared to those with the U haplogroup [3].

Mitochondrial retrograde signaling (from mitochondria to nucleus) has become a major topic for those interested in intracellular communication. Recent studies show that mtDNA variants can modulate gene expression of the complement, inflammation, angiogenesis, and methylation pathways [2,4,5,6,7]. Accumulation of mitochondrial damage and dysfunction is a major feature associated with cancers (e.g., leukemias, glioblastomas, oncocytomas) and age-related diseases (e.g., Alzheimer’s disease, Parkinson’s disease and age-related macular degeneration), suggesting that the normal cellular homeostasis would be affected greatly by mtDNA dysfunctions found in diseased cells [4,8,9,10,11].

Recent studies have used transmitochondrial cybrids (cytoplasmic hybrids) to show that the mtDNA haplogroup profiles can modulate whether cells use predominantly OXPHOS or glycolysis. Studies using human retinal pigment epithelial (RPE) cybrids (cell lines with identical nuclei but mitochondria from either H or J haplogroup individuals) have shown that H cybrids use predominately OXPHOS, produce higher levels of ATP and reactive oxygen species (ROS), and have higher expression levels of mitochondrial encoded genes, including those related to OXPHOS [4,7,8]. Additionally, there are differential expression levels of genes related to apoptosis, complement, and inflammation pathways. Although the nuclear genome and culture conditions were identical, the cybrid cells with J haplogroup mitochondria showed increased rates of cell viability, along with higher glycolysis and lactate production [4,7,12]. These features are also associated with the Warburg effect that is seen in cancer cells [9]. Moreover, the epigenetic profiles of the H and J cybrids were different from each other, indicating mtDNA can modulate the methylation status of cells. The J cybrids showed down-regulation of many genes relating to epigenetic modification including *HDAC1, HAT1, MBD2*, and *DNMT* genes. In addition, the J cybrids had higher levels of total global methylation when compared to the H cybrids [2,6,11].

Our previous study by Patel et al. tested the differential characteristics of the ARPE19-H and ARPE19-J cybrids in response to cisplatin [11]. Their results showed that in addition to the differential qualities expressed under basal untreated conditions, H and J cybrids reacted differently to cisplatin. J cybrids were more sensitive to cisplatin in a cell viability assay and showed significant decreases in mitochondrial membrane potential and ROS levels when treated with cisplatin [11]. In contrast, H cybrids showed no significant change in mitochondrial membrane potential or ROS levels and were less affected by cisplatin in the cell viability assay [11]. Retinal pigment epithelial cybrids with the J haplogroup (RPE-J cybrids) showed preferential use of glycolysis, lower ATP, higher lactate levels, and different methylation patterns compared to ARPE-H cybrids [6,11].

These findings prompted us to expand our understanding of the role of retrograde signaling in cancer cells. Sripathi et al. reported that retinal pigment epithelial cells undergo similar changes to mammary epithelial cells during epithelial mesenchymal transition [13]. Sourisseau et al. stated that breast epithelial and retinal pigment epithelial cells originated from ectoderm tissue [14]. These published studies provide justification for comparing ARPE cybrids to cybrids generated from epithelial cancer cells, such as breast cancer cells.

Our cybrid studies, using cell lines with identical ARPE19 nuclei, [11,15] demonstrated the impact of mtDNA variations in individuals of European, African, Hispanic, and Asian maternal ancestry after cisplatin treatment. These reports have linked an individual’s mtDNA background to differences in their reaction to cisplatin treatment, with retrograde signaling impacting the efficacy and severity of adverse effects. These findings motivated us to investigate the effects of cisplatin on cybrids formed from cancerous cells.

In the present study, MCF7 cells (breast cancer cell line) were converted into a *Rho0* state (lacking mtDNA) and then the mitochondria from the same individuals used in the ARPE-cybrid study were fused with the MCF7-*Rho0* cells to create cybrids. The working hypothesis was since the MCF7 cybrids contained the mitochondria from the same subjects as the ARPE cybrids, the MCF7-J cybrids would exhibit differential properties compared to the MCF7-H cybrids, similar to the non-pathologic RPE-J versus RPE-H cybrids, which would have implications in personalized treatment of cancer patients with respect to cisplatin.

## 2. Results

### 2.1. Cellular Metabolism and Survival in Cisplatin-Treated MCF&-H and MCF7-J Cybrids

We initiated our study by measuring the IC_50_ values for cisplatin in order to determine the concentration of cisplatin required to inhibit cell survival by 50% in MCF7-H and MCF7-J cybrids (Figure 1A,B). The R^2^ value for the MCF7-H cybrids was 0.8851 and the IC_50_ value was 32.62 μm (95% confidence interval 27.02 to 39.37 μM). MCF7-J cybrids had an R^2^ value of 0.94938 and an IC_50_ of 24.25 μM (95% confidence interval: 22.05 to 26.66 μM).

Then, we assessed the survival of MCF7-H and -J cybrids treated with 20, and 40 μM cisplatin. Using the MTT assay, we determined the levels of metabolic activity indicative of cell viability. Without treatment, MCF7-J cybrids grew at similar rates as the MCF7-H cybrids (102% ± 2% versus 100% ± 2%, *p* = 0.45, Figure 1C). Cell survival of MCF7-H cybrids reduced by 13% (*p* < 0.0001) following treatment with 20 μM cisplatin but decreased by 34% (*p* < 0.0001) after treatment with 40 μM cisplatin. Similarly, cell survival decreased to 23% (*p* < 0.0001) and 41% (*p* < 0.0001) for MCF7-J cybrids treated with 20 μM and 40 μM cisplatin, respectively, as compared to untreated MCF7-J cybrids (Figure 1C). When both MCF7-J and MCF7-H cybrids were treated at 20 or 40 µM, there were no significant changes in viability of the MCF7-J compared to MCF7-H cybrids. Thus, cell survival in the MCF7-H and MCF-J cybrids decreased to a similar extent after cisplatin treatment.

### 2.2. Reactive Oxygen Species (ROS) Production in MCF7-H and MCF7-J Cybrids after Cisplatin Treatment

Next, we wanted to investigate if the effect of cisplatin on the cell survival of respective MCF7 cybrids was due to increased ROS levels. The ROS levels, measured in relative fluorescent units (RFU), were decreased in the MCF7-H and MCF7-J after cisplatin treatments (Figure 2A). There was no difference between the untreated MCF7-H cybrids and untreated MCF7-J cybrids after the 48-h incubation period (*p* = 0.4271). The 20 µM and 40 µM cisplatin-treated MCF7-H cybrids showed significantly lower ROS compared to the untreated MCF7-H cybrids (74% ± 2%, *p* < 0.0001; 51% ± 3%, *p* < 0.0001) compared to untreated MCF7-H cybrids (100% ± 3%). The ROS levels also decreased in the 20 µM cisplatin-treated MCF7-J cybrids (64% ± 4%, *p* < 0.0001) and 40 µM cisplatin-treated MCF7-J cybrids (38% ± 4%, *p* < 0.0001) versus the untreated MCF7-J cybrids (95% ± 5%). Since the ROS production levels were normalized to cell survival for each condition, our findings also showed that after cisplatin treatment, the J cybrid cultures showed significantly less ROS production compared to H cybrid cultures (20 µM: *p* = 0.0005; 40 µM: *p* = 0.0011), although the overall pattern is similar in all other respects.

### 2.3. Mitochondrial Membrane Potential (ΔΨm) in MCF&-H and MCF7-J Cybrids after Cisplatin Treatment

We then wanted to establish whether the decrease in ROS levels was due to elevated mitochondrial activity. To investigate this, we determined the level of mitochondrial membrane potential in response to cisplatin treatment in the different MCF7 cybrids. The ΔΨm was not changed after 48 h of incubation at either 20 µM or 40 µM cisplatin in the MCF7-H and MCF7-J cybrids compared to untreated cybrids (Figure 2B). Specifically, the cisplatin-treated MCF7-H cybrids showed similar ΔΨm compared to the untreated H cybrids (102% ± 7% at 20 µM and 110% ± 14% at 40 µM compared to the untreated cybrid at 100% ± 5%, *p* = 0.80 and *p* = 0.49, respectively). The untreated MCF7-J cybrids had similar levels of ΔΨm (106% ± 6%) as the MCF7-H cybrid (*p* = 0.48). After 20 µM and 40 µM cisplatin treatments, MCF7-J cybrids showed similar ΔΨm values (100% ± 7% at 20 µM and 93% ± 9% at 40 µM compared to the untreated MCF7-J cybrids, *p* = 0.52 and *p* = 0.28, respectively). These findings indicate that ΔΨm was not changed by cisplatin treatment in either the MCF7-H or MCF7-J cybrids.

### 2.4. ATP and Lactate Production in MCF7-H and MCF7-J Cybrids

Next, we determined whether MCF7-H and -J cybrids have distinct metabolic profiles. To study this phenomenon, we measured the concentrations of ATP and lactate in the respective cybrids. The MCF7-H and MCF7-J cybrids were cultured for 24 h and the relative amounts of ATP were measured (Figure 2C). Similar levels of ATP production were found in the MCF7-H (100% ± 4.7%) and the MCF7-J cybrids (92.9% ± 3%; *p* = 0.21). The lactate levels were measured in both the 1:2 dilution and the 1:4 dilution samples (Figure 2D). At the 1:2 ratio, the MCF7-H cybrids showed a lactate concentration of 0.25 ± 0.02 mM compared to the MCF7-J cybrids with 0.28 ± 0.02 mM, *p* = 0.17. The 1:4 ratio samples demonstrated the MCF7-H and MCF7-J cybrids had similar lactate levels (0.20 ± 0.01 mM and 0.22 ± 0.01 mM, respectively, *p* = 0.28). Our findings indicate that the MCF7-H and MCF7-J cybrids showed similar ATP and lactate levels suggesting that the H versus J mtDNA haplogroups were not influencing the mode of energy production.

### 2.5. Bioenergetic Profile by Seahorse Flux Analyses

We then determined whether MCF7-derived H and J cybrids have distinct bioenergetic profiles. The bioenergetic profiles for the MCF7-H and MCF7-J cybrids were measured by Seahorse XF Extracellular Flux Analyzer after 24 h of incubation (Figure 3A). The ATP turnover rates for MCF7-H and MCF7-J cybrids were not significantly different from each other (41.1 ± 2.7 and 45.0 ± 3.6 pmol/min/µg, respectively, *p* = 0.42, Figure 3B). The proton leak rates were 15.5 ± 0.9 pmol/min/µg for the MCF7-H cybrids and 16.0 ± 0.6 pmol/min/µg, for the MCF7-J cybrids, *p* = 0.66 (Figure 3C). The spare respiratory capacity for the cybrids were also similar to each other (MCF7-H, 52.5 ± 15 pmol/min/µg and MCF7-J, 47.5 ± 5.1 pmol/min/µg, *p* = 0.77, Figure 3D). Finally, the OCR/ECAR ratios were also not significantly different from each other (MCF7-H, 10.20 ± 0.734 and MCF7-J, 11.17 ± 0.60, *p* = 0.33, Figure 3E).

### 2.6. Gene Expression for the MCF7-J and MCF7-H Cybrids after Cisplatin Treatment

Next, we want to examine the differential gene expression levels associated with cellular stress in response to cisplatin treatment in the cybrids derived from MCF7 and ARPE cells. One of the critical pathways for assessing cellular stress is the accumulation of mitochondrial DNA (mtDNA), which is regulated by the cGAS-STING signaling axis. This axis disrupts mitochondrial homeostasis and mtDNA efflux into the cytosol, activating the IRF3 or NF-B pathway in the process [16,17,18]. Therefore, we proceed by assessing the relative expression levels of the cGAS-STING genes in cybrids generated from MCF7 and ARPE cells in response to cisplatin treatment.

#### 2.6.1. STING-Related Gene Expression

For the *STING*-related genes, the cisplatin-treated MCF-J cybrids showed significant increases for *cGAS* (2.23 ± 0.34-fold, *p* = 0.02) and *IRF3* (1.91 ± 0.46-fold, *p* = 0.04) compared to the untreated J cybrids (Table 1, Figure 4). There were no changes in expression levels for the *STING, TBK1, NFkB2,* and *IFN-a* genes with cisplatin-treated MCF7 cybrids.

In response to cisplatin treatment, the ARPE19-H cybrids showed a significant increase in *cGAS* compared to untreated H cybrids (1.29 ± 0.07-fold, *p* = 0.011), *IRF3* (2.2 ± 0.38-fold, *p* = 0.0006) and *IFNα* (2.21 ± 0.57-fold, *p* = 0.02) (Table 1). The cisplatin-treated ARPE19-J cybrids had upregulation in four genes compared to the untreated ARPE19-J cybrids: *cGAS* (1.78 ± 0.19-fold, *p* = 0.03), *STING* (2.70 ± 0.78-fold, *p* = 0.007), *TBK1* (2.06 ± 0.49-fold, *p* = 0.0004), and *NFkB2* (1.89 ± 0.47-fold, *p* = 0.04).

Following that, we examined the differential expression of cancer-related genes in cybrids generated from the ARPE and MCF7 cell lines in response to cisplatin treatment.

#### 2.6.2. Cancer-Related Gene Expression

The expression patterns for seven cancer-related genes, three apoptosis genes, one signaling gene, and six *STING*-related genes were analyzed for the MCF7 cybrids before and after cisplatin treatment (Table 1, Figure 4). The *ABCC1* expression levels were increased significantly for both treated MCF-H and MCF-J cybrids compared to the untreated cybrids (5.34 ± 1.44-fold, *p* = 0.0001 and 4.61 ± 1.43-fold, *p* = 0.013, respectively). In addition, the *BRCA1* expression level was higher in the untreated MCF7-J cybrids (1.56 ± 0.21-fold, *p* = 0.008) compared to the untreated MCF7-H cybrids (assigned a value of 1). After cisplatin treatment, the MCF7-H cybrids increased *BRCA1* expression over three-fold (3.15 ± 0.78-fold, *p* = 0.019) compared to the untreated MCF7-H cybrid. The cisplatin-treated MCF7-J cybrids also showed higher *BRCA1* expression (2.78 ± 0.48-fold, *p* = 0.009) compared to the untreated MCF7-J cybrid.

In contrast, decreased transcription levels were seen for the *EGFR* gene after cisplatin treatment. The cisplatin-treated MCF7-H cybrids and MCF7-J cybrids had lower *EGFR* levels (0.53 ± 0.11-fold, *p* = 0.036 and 0.45 ± 0.13-fold, *p* = 0.05) as compared to the untreated H cybrids and J cybrids. The treated MCF7-H and MCF7-J showed a 5.99-fold (*p* = 0.0023) and 2.97-fold (*p* = 0.0049) increase in *CDKN1A/P21*, respectively. There were no differences in expression levels for *CYP51A*, *DHRS2/HEP27, ERBB2,* and *ERCC1* after cisplatin treatment. Furthermore, there were negligible amounts of RNA expression for *ALK1* in the MCF7 cybrids. The ARPE-19-H and ARPE-19-J cybrids showed differential expression of two cancer-related genes after cisplatin treatment (Table 1, Figure 4). The cisplatin-treated ARPE19-H cybrids showed a significant increase in *CDKN1A/P21* expression compared to untreated ARPE19-H cybrids (4.89 ± 0.51-fold, *p* = 0.002). *CYP51A* expression was higher in the ARPE19-J treated cybrids (1.94 ± 0.12-fold, *p* = 0.0057) compared to untreated, while the APRE19-H treated cybrids were not altered (*p* = 0.12).

## 3. Discussion

There are many studies that have demonstrated crosstalk between the mitochondria and nuclear gene expression [6,11,19,20]. The importance of this retrograde signaling has become clearer using the transmitochondrial cybrid model where all cell lines have identical nuclei but contain mitochondria from individuals of different mitochondrial haplogroups (i.e., H and J cybrids). Our previous studies comparing the ARPE19-H cybrids versus the ARPE19-J cybrids showed that the retrograde signaling from different haplogroups modulated the different cell lines differently [4,6,11], which made them of great interest in cancer studies. The ARPE19-H cybrids preferentially used OXPHOS, had higher levels of ROS and ATP production, lower levels of lactate, and increased rates of cell growth compared to the J cybrids [11,21]. When mitochondria from the identical subjects were placed into the MCF7 Rho*0* cells, the responses were significantly different than those seen in the ARPE19 cybrids. The MCF7-H and MCF7-J cybrids had similar levels of ATP, lactate, ROS formation, mitochondrial membrane potential, OCR:ECAR ratios, and Seahorse^®^ bioenergetic values. These findings suggest that the nuclei in ARPE19 cybrids are receptive to the retrograde signaling originating from the H and J mitochondria. In contrast, the MCF7 nuclei are resistant to the retrograde signaling and therefore, the MCF7-H and MCF7-J cybrids exhibit similar behaviors (Figure 5).

It has been documented that the cGAS-STING pathway is capable of sensing damaged mitochondria through cytosolic mtDNA and activation following this signal. Therefore, this pathway was determined to be possibly responsible for the retrograde signaling that was seen in this study. When expression levels of STING genes were measured, the cisplatin-treated ARPE19-H and ARPE19-J cybrids showed significant increases in expression levels of seven STING genes, including the downstream *IFN-a*, similar to the results found in the study from Parkes, et al. [22]. After cisplatin treatment, the responses of the ARPE19 cybrids differed from each other with increases for the *STING, TBK1*, and *NFkB1* expression in ARPE19-J-treated cells and higher levels of *IFN-α* and *IRF3* in the ARPE19-H-treated cybrids compared to their respective untreated controls. Since all cybrid cell lines had identical nuclear genomes, the differential responses of the STING pathway genes are due to the modulation by the H versus J mtDNA haplogroups. In contrast, the cisplatin-treated MCF7-H cybrids did not show any significant changes in STING genes and the cisplatin-treated MCF7-J cybrids showed upregulation for only the *cGAS* and *IRF3* genes. This leads to the conclusion that the MCF7 cybrids failed to activate its STING pathway in the presence of s-phase-specific DNA damage caused by cisplatin, which would normally be expected [22]. While the exact mechanism behind this is unclear, it could be hypothesized that the cGAS-STING cytosolic DNA sensing system is interfered with by the cancerous nucleus, obstructing the cGAS-STING response that would normally follow cisplatin-induced damage. Another possibility may be that the DNA repair mechanisms inherent in the MCF7 cell line are robust enough that minimal cytosolic DNA is produced from cisplatin-induced damage, eliminating the need for the cGAS-STING response.

In the untreated MCF7 cybrids, the expression levels for the *BRCA1* gene were significantly higher in the MCF7-J cybrids compared to the MCF7-H cybrids (*p* = 0.0076). If similar responses were found in cancer patients, then individuals with the mtDNA J haplogroup might have higher levels of *BRCA1* that might provide protection since it maintains genomic stability by repairing DNA breaks and yields tumor suppression. Moreover, after cisplatin treatment, the MCF7-H and MCF7-J cybrids showed 3.15-fold and 2.78-fold increase in *BRCA1* transcription, respectively, which may contribute to the inhibition of tumor growth from the treatment. Although the ARPE19 and MCF7 cybrids contained mitochondria from identical subjects, the responses to cisplatin were different in the MCF7 cybrids compared to the ARPE cybrids [6]. After cisplatin treatment, the cisplatin-treated ARPE19 cybrids showed differential expression for *CDKN1A/P21* (ARPE19-H was upregulated, ARPE19-J had no change) and *CYP51A* (ARPE19-J was upregulated, APRE19-H showed no change). There were no significant changes in gene expression levels for the seven other cancer genes (*ABCC1*, *ALK1, BRCA1, DHRS2/HEP27*, *EGFR*, *ERBB2*, and *ERCC1*). In contrast, both MCF7-H and MCF7-J cybrids showed parallel levels of upregulation in *ABCC1*, *BRCA1*, and *CDKN1A/P21*, along with lower transcription of the *EGFR* gene.

After cisplatin treatment, the MCF7-H and MCF7-J cybrids exhibited a five-fold and four-fold upregulation of the *ABCC1* gene, respectively, which is also known as “multidrug resistance protein 1” and has been associated with cisplatin resistance. Tommasi S et al., reported that the H haplogroup was associated with breast cancer tumor grade differentiation and with *BRCA2* mutations [23], suggesting that mtDNA variants may act as a genetic modifier for breast cancer. While the mtDNA U haplogroup has been overrepresented in control subjects [24], the mtDNA I haplogroup has higher association with the breast cancer group [25], although there is still some controversy of the association between European mtDNA haplogroups and breast cancer risk [26]. Multiple studies have speculated that in Asian, African-American, and European-American populations, there may be an increased association between the m.10398G > A polymorphism and breast cancer [27,28,29,30]. The m.10398G > A is a non-synonymous SNP in the *MT-ND3* gene, and it has been speculated that this alternation in Complex I would result in higher ROS production and increased apoptosis of normal cells, thereby allowing proliferation of cancer cells.

Another Chinese study using cybrids created in MDAMB-231 cells showed those with the D5 haplogroup mtDNA had lower mitochondrial respiration, ATP production, and mitochondrial membrane potential along with tumorigenic behavior compared to non-D5 cybrids [31]. The difference in these studies showing that the D5 haplogroup modulated the cell behavior which was absent in our MCF7 cells may be due to the fact that the MDAMB-231 cells are metastatic mammary adenocarcinoma 1 while the MCF7 cell line is also from a metastatic differentiated mammary epithelial cell that can possess estrogen receptors and can process estradiol.

## 4. Methods

### 4.1. ARPE19-H, ARPE19-J, MCF7-H, and MCF7-J Cybrids

All subjects read and signed an informed consent (IRB #2003-3131) from the Institutional Review Board of the University of California, Irvine. All clinical investigations and protocols were conducted according to the principles of the Declaration of Helsinki and approved by the appropriate investigational review boards (University of California, Irvine). Cybrids were generated as described previously [4,6,12]. H and J cybrids were created by polyethylene glycol fusion of platelets with the *Rho**0* (mtDNA free) ARPE-19 or MCF-7 cells, which had been treated by low dosage ethidium bromide as described by Miceli [19]. H and J cybrids were cultured to the fifth passage using DMEM-F12 containing 10% dialyzed fetal bovine serum, 100 unit/mL penicillin, 100 µg/mL streptomycin, 2.5 μg/mL fungizone, 50 µg/mL gentamycin, and 17.5 mM glucose. Cybrid haplogroups were verified by PCR, restriction enzyme digestion, and/or mtDNA sequencing. According to PhyloTree.org, the J haplogroup was defined by the SNPs at m.295C > T, m.489T > C, m.12612A > G, m.13708G > A, and m.16069C > T. The specific J cybrids used in this study included those with haplogroups J1d1a, J1c1, J1b1a, and J1c1a. As for the H cybrids, the cell lines included those with the H4a1a4b2, H1e1a, H1j, and H1 haplogroups. The defining SNPs for the H haplogroup according to PhyloTree.org were m.7028T > C and m.2706G > A. Further specific haplotyping was also done according to the haplotyping tree on PhyloTree.org.

### 4.2. IC_50_ Analysis of Cisplatin Titration Curve Measuring Cell Survival

MCF7-H and MCF7-J cybrids were plated in 96-well plates (10,000 cells/well), incubated for 24 h, and then treated with 0, 20, 40, 60, 80, 100, or 120 µM of cisplatin. The cybrids were incubated for another 48 h before having their cell viabilities measured with an MTT reagent (3-(4,5-dimethylthiazol-2-yl)-2,5-diphenyltetrazolium bromide) (Cat. # 30006, Biotium, Inc., Fremont, CA, USA). In this assay, the cybrids were incubated for two hours with the MTT reagent and absorbance was measured with the ELx808 spectrophotometer (BioTek Instruments/Agilent, Winooski, VT, USA) at 570 nm with reference wavelength at 630 nm. The background absorbance was subtracted from the signal absorbance and values were normalized to the untreated specimen of each cell line. Each treatment was analyzed with eight replicates. An IC_50_ analysis was done to determine the concentration of cisplatin required to inhibit the cell viability by 50% (GraphPad Prism Software, Inc., Version 5.0, San Diego, CA, USA).

### 4.3. Cell Survival Assay

ARPE19-H, ARPE19-J, MCF7-H, and MCF7-J cybrids were plated in 96-well plates (10,000 cells/well) and incubated for 24 h. Then cisplatin was added to the media at concentrations of 0, 20, or 40µM and incubated for another 48 h. MTT reagent (Cat. # 30006, Biotium, CA, USA) was added to the cultures for two hours and absorbance was measured using the ELx808 spectrophotometer (BioTek, Winooski, VT, USA) at 570 nm with reference wavelength at 630 nm. The background absorbance was subtracted from the signal absorbance and values were normalized to H untreated and analyzed using the two tailed *t*-test (GraphPad Software Inc., San Diego, CA, USA). Experiments analyzed in triplicate and the entire experiment was repeated twice.

### 4.4. Reactive Oxygen Species (ROS) Assay

ARPE19-H, ARPE10-J, MCF7-H, and MCF7-J cybrids were plated in 96-well plates (10,000 cells/well) and incubated for 24 h. Cells were treated with 0, 20, or 40 µM of cisplatin for another 48 h. ROS levels were measured with fluorescent dye 2,7-dichlorodihydrofluorescin diacetate (H_2_DCFDA, Invitrogen-Molecular Probes, Carlsbad, CA, USA) on a fluorescence plate reader using 490 nm for emission and 520 nm for excitation wavelengths (Gemini XPS Microplate Reader, Molecular Devices, Sunnyvale, CA, USA).

The data results were normalized to the untreated MCF-H group and normalized to their respective cell viability values to account for cell number differences. Differences in cisplatin-treated cells compared to untreated cells were analyzed (Prism, GraphPad Software Inc., San Diego, CA, USA) and were considered to be statistically significant when *p* ≤ 0.05. Experiments were analyzed in quadruplicate and the entire experiment repeated three separate times.

### 4.5. Mitochondrial Membrane Potential (ΔΨm) Assay

ARPE19-H, ARPE19-J, MCF7-H, and MCF7-J cybrids were plated in 96-well plates (10,000 cells/well), incubated for 24 h, and treated with 0, 20, or 40 µM of cisplatin for another 48 h. JC-1 reagent (5,5′,6,6′-tetrachloro-1,1′,3,3′- tetraethylbenzimidazolylcarbocyanine iodide) (Biotium, Hayward, CA, USA) was added to cultures for 15 min. Fluorescence was measured using the Gemini XPS Microplate Reader (Molecular Devices) for red (excitation 550 nm and emission 600 nm) and green (excitation 485 nm and emission 545 mm) wavelengths. Intact mitochondria with normal ΔΨm appeared red, while cells with decreased ΔΨm were in a green, fluorescent state. Experiments were analyzed in quadruplicate and the entire experiment was repeated three separate times. Cisplatin-treated values were compared to untreated values for statistical significance (*p* ≤ 0.05, GraphPad Prism Software, Inc. Version 5.0, San Diego, CA, USA).

### 4.6. ATP Assay

ARPE19-H, ARPE19-J, MCF7-H, and MCF7-J cybrids were plated in 96-well plates (10,000 cells/well) and incubated for 24 h. The cells were then lysed with a lysis solution and lightly shaken for five minutes. A substrate solution containing luciferase and luciferin (Abcam, Cambridge, UK) was then added, and the cells were lightly shaken for another five minutes. The cells were then stored in the dark for 10 min before the luminescence was read using the Gemini XPS Microplate Reader (Molecular Devices, San Jose, CA, USA) at 560 nm. Values were normalized to H untreated and analyzed using the two tailed T-test (Prism, GraphPad Software Inc., Version 5.0, San Diego, CA, USA). Experiments were analyzed in triplicate and the entire experiment was repeated twice.

### 4.7. Lactate Assay

The lactate levels were measured using the Lactate Assay Kit (Eton Bioscience Inc., San Diego, CA, USA) and according to the manufacturer’s protocol. MCF7-H and MCF7-J cybrids were plated at 10,000 cells/well in 96-well plates, incubated overnight and lactate concentrations were analyzed. The solutions were diluted 1:2 and 1:4 to verify the repeatability of the assay. Standards and samples were set up as duplicates and quadruplicates and experiments were repeated twice.

### 4.8. Seahorse Extracellular Flux Analysis

Bioenergetic profiles for the cybrids were measured by Seahorse XF Extracellular Flux Analyzer (Agilent, Santa Clara, CA, USA). MCF7-H (*n* = 3) and MCF7-J (*n* = 3) cybrids were plated at 50,000 cells/well and cultured overnight at 37 °C under 5% CO_2_. Samples were run in triplicate and experiments repeated three times. Plates were then washed and placed for 1 h in a 37 °C incubator under air in 500 μL of unbuffered DMEM (Dulbecco’s modified Eagle’s medium, pH 7.4), supplemented with 17.5 mM glucose (Sigma, St Louis, MO, USA), 200 mM L-glutamine (Invitrogen-Molecular Probes, Carlsbad, CA, USA), and 10 mM sodium pyruvate (Invitrogen Molecular Probes, Carlsbad, CA, USA). There were sequential injections into the wells of oligomycin (1 μM final concentration, which blocks ATP synthase to assess respiration required for ATP turnover), FCCP (1 μM final concentration, a proton ionophore which induces chemical uncoupling and maximal respiration), and rotenone plus antimycin A (1 μM final concentration of each, completely inhibits electron transport to measure non-mitochondrial respiration). Data from each well were normalized by measuring the total protein. Total protein was isolated using RIPA lysis buffer (Millipore, Billerica, MA, USA) containing protease inhibitor (Sigma, St. Louis, MO, USA) and phosphatase arrest (G biosciences, St. Louis, MO, USA). Isolated protein was then mixed with Qubit buffer and measured with Qubit 2.0 fluorometer (Invitrogen, Grand Island, NY, USA).

All data from XF24 assays were collected using the XF Reader software from Seahorse Bioscience. The oxygen consumption rate (OCR) was determined by measuring the drop in O_2_ partial pressure over time followed by linear regression to find the slope. The extracellular acidification rate (ECAR) was determined by measuring the change in pH levels over time followed by linear regression to find the slope of the line, which represents ECAR. The percentage ATP Turnover Rate is calculated by the following formula: 100 − (ATP coupler response/basal respiration × 100). The percentage Spare Respiratory Capacity represents a bioenergetic value for cells needing high amounts of ATP in response to demands placed upon them. This is calculated by the formula: Electron transport chain (ETC) accelerator response/basal respiration × 100. The percentage Proton Leak equals the ATP coupler response—non-mitochondrial respiration. Data from these experiments were exported to GraphPad Prism 5 (GraphPad Software, Version 5.0, San Diego, CA, USA) where they were analyzed, normalized, and graphed. Statistical significance was determined by performing two-tailed Student t tests and *p* ≤ 0.05 was considered significant in all experiments.

### 4.9. RNA Isolation, cDNA Synthesis and Quantitative Reverse Transcription PCR (qRT-PCR)

ARPE19-H, ARPE19-J, MCF7-H, and MCF7-J cells were plated (500,000 cells/well) and incubated for 24 h in six-well plates. MCF-7 cybrids were treated with culture media containing either 0 or 20 μM of cisplatin for another 48 h. Trypsinized cells were pelleted, and RNA isolated following the manufacturer’s protocol (PureLink RNA Mini Kit, Invitrogen, Carlsbad, CA, USA). After RNA quantification (Nanodrop 1000, Thermoscientific, Wilmington, DE, USA), the cDNA was transcribed from 100 ng of RNA (SuperScript IV VILO, Invitrogen), and then used for quantitative reverse transcription-PCR (qRT-PCR) (QuantStudio3 instrument; Applied Biosystems, Carlsbad, CA, USA). SYBR Green-based primers were used (Qiagen, Redwood City, CA, USA).

Table 2 shows the GenBank Accession numbers and functions for 18 genes that were investigated. The six genes, related to the STING pathway, are cyclic GMP-AMP synthase (*cGAS*), stimulator of interferon genes (*STING*), TANK binding kinase 1 (*TBK1*), interferon regulatory factor 3 (*IRF3*), nuclear factor kappa B subunit 2 (*NFkB2*), and interferon alpha (*IFN-a*). The nine cancer-related genes were excision repair cross-complementation group 1 (*ERCC1*), dehydrogenase/reductase member 2 (*DHRS2/HEP27*), ATP-binding cassette, sub-family C (*ABCC1*), breast cancer type 1 susceptibility protein (*BRCA1*), epidermal growth factor receptor 2 (*ERBB2*), activin receptor-like kinase 1 (*ALK1*), and epidermal growth factor receptor (*EGFR*), cyclin-dependent kinase inhibitor 1A/p21 (*CDKN1A/P21*) and cytochrome P450, family 51, subfamily A, polypeptide 1 (*CYP51A*). We also examined BCL2-associated X protein (*BAX*), caspase-3 (*CASP3*), and caspase-9 (*CASP9*), which show pro-apoptosis. Target cycle thresholds (Ct) values were initially compared to the Ct values of reference genes and subsequently, comparisons between untreated and cisplatin-treated values (ΔΔCt) were evaluated for statistical significance. Fold differences were calculated using the equation 2^(ΔΔCt)^.

### 4.10. Statistical Analyses

Statistical analysis of the data was performed by ANOVA (GraphPad Prism, version 5.0). Newman–Keuls multiple-comparison or the two-tailed *t*-tests were used to compare the data within each experiment. *p* < 0.05 was considered statistically significant. Error bars in the graphs represent standard error of the mean (SEM).

## 5. Conclusions

To conclude, this is the first study showing the altered pattern of the retrograde signaling pathway in MCF7-H and MCF-J mitochondrial cybrids. Our findings support that the retrograde signaling processes may differ depending upon the specific mtDNA haplogroup within the cells. Moreover, the non-cancerous nucleus may respond differently to the retrograde mitochondrial signals than the nuclei within the cancer cells. This area of research requires additional investigation but has great potential to identify novel mechanisms of mitochondrial-nuclear interactions and new pathways that have not yet been identified.

## Figures and Tables

**Figure 1 ijms-23-06687-f001:**
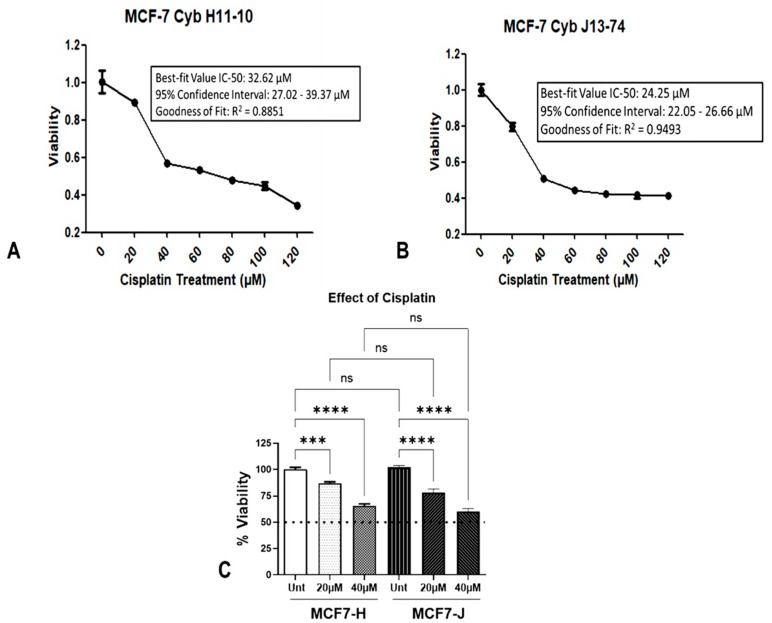
Effect of cisplatin on the survival of cybrids derived from MCF7 with distinct mtDNA haplogroups. (**A**) H and (**B**) J cybrids were plated in 96-well plates, cultured for 24 h prior to the addition of cisplatin (0, 20, 40, 60, 80, 100, or 120 µM) and then incubated for another 48 h. Cell viabilities of the cybrids were measured using the colorimetric MTT assay and normalized to the untreated MCF7-H cybrid sample. An IC-50 analysis was done to determine the concentration at which viability decreased by 50%. For the MCF7-H cybrids the IC-50 was 32.62 µM (R^2^ = 0.8851) and the MCF7-J cybrid had a value of 24.25 µM (R^2^ = 0.9493). (**C**) Both the MCF7-H and MCF7-J cybrids had significant but similar decreases in survival in response to cisplatin. At 20 µM of cisplatin, viability for the MCF7-H and MCF7-J cybrids decreased by 13% (**** *p* < 0.0001) and 22% (**** *p* < 0.0001) and (*** *p* <0.001), respectively. There was also a significant decrease in response to 40 µM of cisplatin of 34% (*p* < 0.0001) and 40% (*p* < 0.0001) for MCF7-H and MCF7-J cybrids respectively. Each MCF7-H and MCF7-J haplogroup had four different biological cell lines each, with 12 samples for each treatment and cell line group.

**Figure 2 ijms-23-06687-f002:**
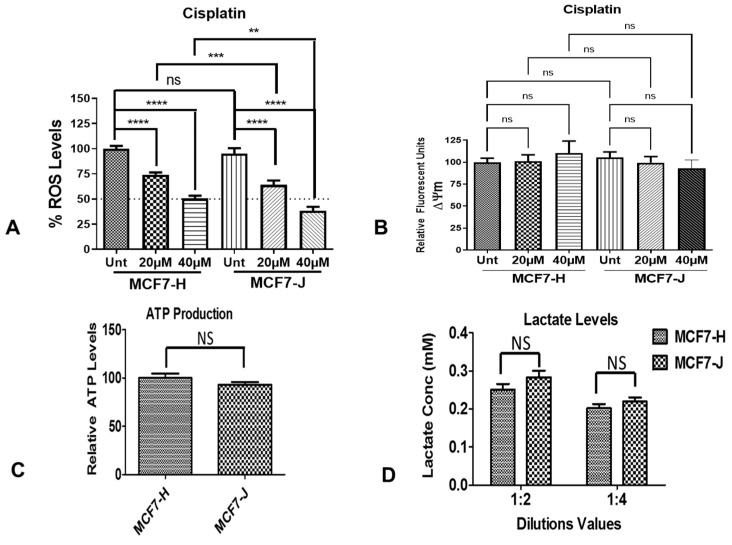
Effect of cisplatin on mitochondrial function of distinct MCF7 cybrids. The H and J MCF-7 cybrids responded similarly to cisplatin. MCF7-H and MCF7-J cybrids were plated in 96-well plates, cultured for 24 h prior to the addition of cisplatin (0, 20, or 40 µM) and then incubated for another 48 h. (**A**) The ROS levels, measured in relative fluorescent units (RFU), were normalized to untreated MCF7-H cybrids. Both the MCF7-H and MCF7-J cybrids had significant decreases in ROS levels in response to 20 µM of cisplatin, 74% (*p* < 0.0001) and 64% (*p* < 0.0001) respectively. There was also a significant decrease in response to 40 µM of cisplatin of 51% (*p* < 0.0001) and 38% (*p* < 0.0001) for MCF7-H and MCF7-J cybrids respectively. Each MCF7-H and MCF7-J haplogroup had four different biological cell lines each, with 12 samples for each treatment and cell line group. (**B**) No significant effect on ΔΨm was found for the MCF-7 H or J cybrids after cisplatin treatment. H and J cybrids were plated in 96-well plates, cultured for 24 h prior to the addition of cisplatin (0, 20, or 40 µM) and then incubated for another 48 h. The ΔΨm values were detected using the JC-1 dye assay kit and the relative fluorescence units (RFU) were normalized to untreated cybrids. For the MCF7-H cybrids, no significant difference was found for the 20 or 40 µM. No significant difference was found for the MCF7-J cybrids at 20 (*p* = 0.52) or 40 µM (*p* = 0.28) as well. Each H and J haplogroup had four different biological cell lines each, with 12 samples for each treatment and cell line group. (**C**) ATP levels in MCF7-H and MCF7-J cybrids were not significantly different (*p* = 0.21). (**D**) Lactate levels in MCF7-H and MCF7-J cybrids showed no significant differences in either dilution experiments. The 1:2 experiment had *p* = 0.17 and the 1:4 experiment had *p* = 0.28. *p*-values: NS denotes non-significant, ** *p* ≤ 0.01; *** *p* ≤ 0.001 and **** *p* ≤ 0.0001.

**Figure 3 ijms-23-06687-f003:**
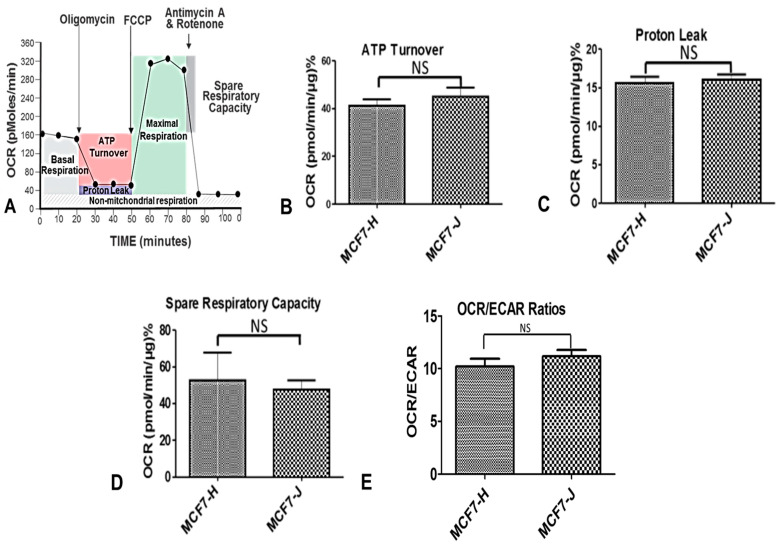
Effect of cisplatin on the bioenergetic profile and oxygen consumption rate (OCR). (**A**) Summary of the Seahorse assay and the different environments it uses to measure OCR for understanding OXPHOS characteristics; MCF7-H and MCF7-J cybrids had no significant differences in (**B**) ATP turnover (*p* = 0.42), (**C**) proton leakage (*p* = 0.66), or (**D**) spare respiratory capacity (*p* = 0.77); (**E**) Under basal conditions, there were also no significant differences in OCR/ECAR (extracellular acidification rate) ratios between the MCF7-H and MCF7-J cybrids (*p* = 0.33).

**Figure 4 ijms-23-06687-f004:**
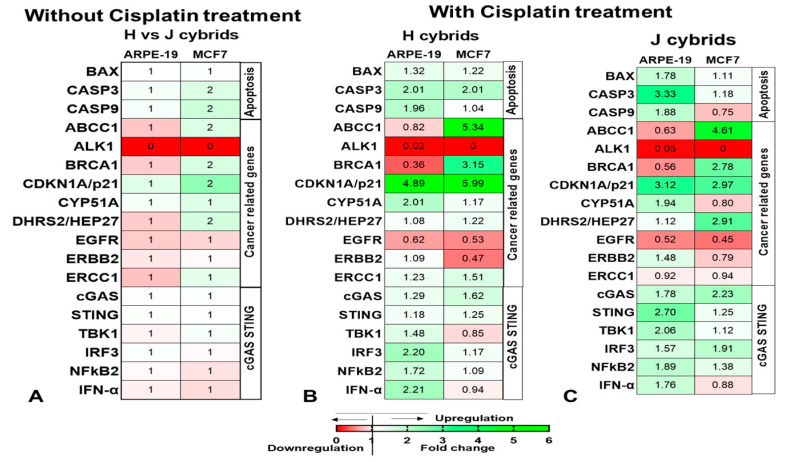
Heatmap depicting the relative expression of all genes related to cancer, cGAS-STING pathway (**A**), and apoptosis pathway in H (**B**) and J (**C**) cybrids formed from ARPE and MCF7 cells in response to cisplatin treatment.

**Figure 5 ijms-23-06687-f005:**
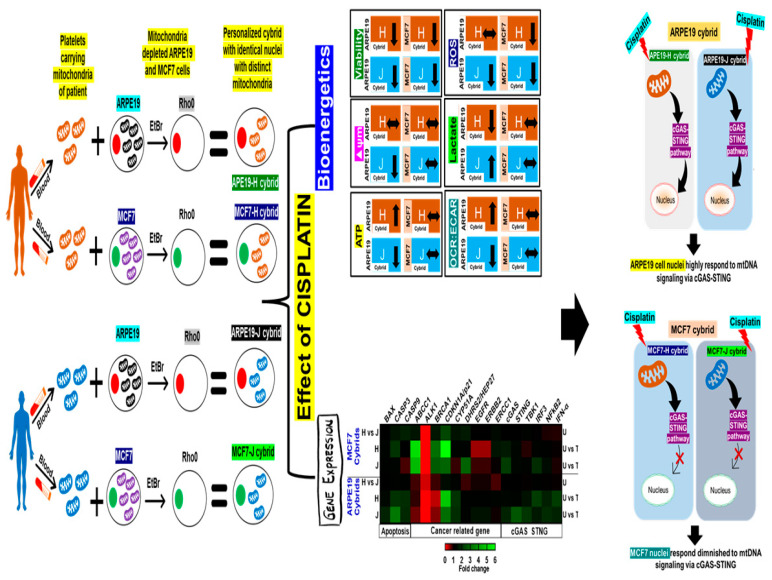
Schematic representation. Summary of MCF7 versus ARPE19 with identical mitochondria.

**Table 1 ijms-23-06687-t001:** Gene expression levels of MCF7-H versus MCF7-J cybrids and ARPE19-H versus ARPE19-J cybrids after treatment with cisplatin.

Gene Symbol	MCF7 Cybrids			ARPE19 Cybrids		
	Untreated Cybrids H# vs. J *p* Value Fold	H Cybrids Untreated# vs. Treated *p* Value Fold	J Cybrids Untreated# vs. Treated *p* Value Fold	Untreated Cybrids H# vs. J *p* Value Fold	H Cybrids Untreated# vs. Treated *p* Value Fold	J Cybrids Untreated# vs. Treated *p* Value Fold
cGAS STING Pathway						
*cGAS*	0.86 1.04 ± 0.23	0.16 1.62 ± 0.41	0.02 2.23 ± 0.34	0.89 1.03 ± 0.24	0.01 1.29 ± 0.07	0.03 1.78 ± 0.19
*STING*	0.95 1.01 ± 0.38	0.24 1.25 ± 0.17	0.18 1.25 ± 0.22	0.89 1.05 ± 0.30	0.47 1.18 ± 0.39	0.007 2.70 ± 0.78
*TBK1*	0.47 1.13 ± 0.19	0.34 0.85 ± 0.13	0.38 1.12 ± 0.17	0.83 0.95 ± 0.05	0.37 1.48 ± 0.33	0.0004 2.06 ± 0.49
*IRF3*	0.93 0.98 ± 0.14	0.48 1.17 ± 0.23	0.04 1.91 ± 0.46	0.83 1.07 ± 0.52	0.0006 2.2 ± 0.38	0.22 1.57 ± 0.34
*NFkB2*	0.76 0.88 ± 0.23	0.84 1.09 ± 0.41	0.16 1.38 ± 0.31	0.89 0.97 ± 0.082	0.149 1.72 ± 0.32	0.04 1.89 ± 0.47
*IFN-a*	0.68 0.86 ± 0.21	0.88 0.94 ± 0.31	0.70 0.88 ± 0.42	0.84 0.94 ± 0.34	0.02 2.21 ± 0.57	0.12 1.76 ± 0.61
CANCER-RELATED						
*ABCC1*	0.27 1.51 ± 0.51	0.0001 5.34 ± 1.44	0.013 4.61 ± 1.43	0.35^^ 0.77 ± 0.19	0.24^^ 0.82 ± 0.078	0.21^^ 0.63 ± 0.095
*ALK1*	Not Expressed	Not Expressed	Not Expressed	0.21 0.29 ± 0.19	0.10 0.02 ± 0.01	0.13 0.05± 0.032
*BRCA1*	0.008 1.56 ± 0.21	0.019 3.15 ± 0.78	0.009 2.78 ± 0.48	0.59 0.79 ± 0.32	0.06 0.36 ± 0.09	0.26 0.56 ± 0.19
*CDKN1A/* *P21*	0.11 2.41 ± 0.68	0.0023 5.99 ± 0.95	0.0049 2.97 ± 0.295	0.61^^ 1.41 ± 0.66	0.002^^ 4.89 ± 0.51	0.12^^ 3.12 ± 0.65
*CYP51A*	0.2 1.495 ± 0.3	0.45 1.17 ± 0.10	0.26 0.797 ± 0.058	0.63^^ 1.14 ± 0.14	0.12^^ 2.01 ± 0.47	0.0057^^ 1.94 ± 0.12
*DHRS2/* *HEP27*	0.36 1.61 ± 1.12	0.83 1.22 ± 1.05	0.41 2.91 ± 2.51	0.47 0.79 ± 0.17	0.89 1.08 ± 0.14	0.75 1.12 ± 0.42
*EGFR*	0.51 0.83 ± 0.22	0.036 0.53 ± 0.11	0.05 0.45 ± 0.13	0.62 0.79 ± 0.37	0.19 0.62± 0.16	0.31 0.52 ± 0.23
*ERBB2*	0.94 0.98 ± 0.31	0.06 0.47 ± 0.17	0.62 0.79 ± 0.37	0.78 0.90 ± 0.31	0.87 1.09 ± 0.55	0.32 1.48 ± 0.45
*ERCC1*	0.24 1.36 ± 0.31	0.37 1.51 ± 0.55	0.84 0.94 ± 0.29	0.08 0.75 ± 0.07	0.32 1.23 ± 0.18	0.41 0.92 ± 0.17
APOPTOSIS						
*BAX*	0.83 1.04 ± 0.186	0.51 1.22 ± 0.33	0.57 1.11 ± 0.21	0.97^^ 1.01 ± 0.09	0.32^^ 1.32 ± 0.26	0.05^^ 1.78 ± 0.26
*CASP3*	0.057 1.52 ± 0.11	0.0167 2.01 ± 0.24	0.22 1.18 ± 0.103	0.63^^ 1.14 ± 0.14	0.12^^ 2.01 ± 0.47	0.02^^ 3.33 ± 0.62
*CASP9*	0.22 1.72 ± 0.32	0.28 1.04 ± 0.16	0.18 0.75 ± 0.045	0.72 1.11 ± 0.29	0.1 1.96 ± 0.36	0.1 1.88 ± 0.27
	MCF7 CYBRIDS			ARPE-19 CYBRIDS		

**Table 2 ijms-23-06687-t002:** Description and function of interested genes.

Symbol	Gene Name	GenBank Accession Numbers	Functions
cGAS-STING pathway			
cGAS	Cyclic GMP-AMP synthase	NM_138441	Catalyzes the formation of cyclic GMP-AMP from ATP and GTP and plays a key role in innate immunity.
STING	Stimulator of interferon genes; transmembrane protein 173	NM_198282	Facilitator of innate immune signaling that acts as a sensor of cytosolic DNA from bacteria and viruses and promotes the production of type I interferon (IFN-alpha and IFN-beta).
TBK1	TANK binding kinase 1	NM_013254, XM_005268809, XM_005268810	Serine/threonine kinase that plays an essential role in regulating inflammatory responses to foreign agents.
IRF3	Interferon regulatory factor 3	NM_001197122, NM_001197123, NM_001197124, NM_001197125, NM_001197126, NM_001197127, NM_001197128, NM_001571, XM_006723197, XM_006723198, XM_017023766, XM_017023767	Key transcriptional regulator of type I interferon (IFN)-dependent immune responses which plays a critical role in the innate immune response against DNA and RNA viruses.
NFkB2	Nuclear factor kappa B subunit 2	NM_001077494, NM_001261403, NM_001288724, NM_001322934, NM_001322935, NM_002502	Transcription factor produced at the endpoint of many signal transduction processes relating to inflammation, immunity, differentiation, cell growth, tumorigenesis, and apoptosis.
IFN-a	Interferon alpha	NM_006900, NM_024013	Stimulates production of a protein kinase and an oligoadenylate synthetase for antiviral activities.
CANCER-RELATED			
ABCC1	ATP-binding cassette, subfamily C	NM_004996	Known as MRP1 (multidrug resistance protein 1). Member of the ATP binding cassette family that transports molecules across membranes. Mutations in ABCC1 N-glycosylation connected with cisplatin resistance.
ALK1	Activin receptor-like kinase	NM_000020	Type I cell-surface receptor for TGF-beta superfamily of ligands. Shares similar domain structures in serine-threonine kinase subdomains with other activin receptor-like kinase proteins.
BRCA1	Breast cancer Type 1 susceptibility protein	NM_007294	Nuclear phosphoprotein that acts as a tumor suppressor by maintaining genomic stability. Involved in transcription, DNA repair of double-stranded breaks, and recombination.
CDKN1A/ P21	Cyclin-dependent kinase inhibitor 1A/p21	NM_000389, NM_00122077, NM_00122077, NM_078467, NM_001291549	Plays a critical role in the cellular response to DNA damage and cisplatin toxicity. Mediates cell cycle arrest. Is cyto-protective.
CYP51A	Cytochrome P450, family 51, subfamily A, polypeptide 1	NM_000786, NM_001146152	Member of the cytochrome P450 enzyme family of monooxygenases.
DHRS2/ HEP27	Dehydrogenase/ reductase (SDR Family) member 2	NM_182908, NM_005794	NADPH-dependent dicarbonyl reductase activity; Mitochondrial matrix protein. Inhibits MDM2 and stabilizes p53.
EGFR	Epidermal growth factor receptor	NM_005228	Triggers cell proliferation when bound to epidermal growth factor.
ERBB2	Erb-b2 receptor tyrosine kinase 2	NM_004448	Member of epidermal growth factor receptor family of receptor tyrosine kinases. Stabilizes binding of epidermal growth factor to receptor.
ERCC1	Excision repair cross-complementation group 1	NM_001166049, NM_001983, NM_202001	Nucleotide excision repair formed by electrophilic compounds such as cisplatin. Forms a heterodimer with XPF endonuclease. Involved in recombination DNA repair, inter-stand crosslink, and lesion repair.
APOPTOSIS			
BAX	BCL2-associated X protein	NM_001291429, NM_001291428, NM_001291430, NM_138761, NM_004324, NM_138764, NM_001291431, NM_138763	Associates and forms a heterodimer with BCL2. Functions in apoptotic behavior by opening the mitochondrial voltage dependent anion channel, leading to loss of membrane potential and opening of cytochrome C.
CASP3	Caspase-3	NM_004346, NM_032991	Effector caspase; Activated by caspases 8, 9 and 10; Effects caspases 6, 7, 9. Belongs to family of proteases involved in apoptosis. Synthesized as inactive precursors and therefore need activation.
CASP9	Caspase-9	NM_001229, NM_032996, XM_005246014	Part of the apoptosome protein complex formed during apoptosis. Mitochondrial caspase activation.
			HOUSEKEEPERS
HPRT1	Hypoxanthine phosphor-ribosyl-transferase 1	NM_000194	Transferase catalyzes conversion of hypoxanthine to inosine monophosphate and guanine to guanosine monophosphate.
HMBS	Hydroxy-methylbilane synthase	NM_000190, NM_001024382, NM_001258208, NM_001258209	Member of the HMBS superfamily. Third enzyme of heme biosynthetic pathway; catalyzes head to tail condensation of four porphobilinogen molecules into the linear hydroxymethylbilane.

GenBank Accession Numbers and Functions from Uniprot.org.

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
