# Peer review of "Altered Retrograde Signaling Patterns in Breast Cancer Cells Cybrids with H and J Mitochondrial DNA Haplogroups"

_ijms, 2022, doi:10.3390/ijms23126687_

Round 1

Reviewer 1 Report

The findings appear to be interesting and technically well performed. Specific points that the authors need to address are as follows:

1. Most of the experiments have been done in MCF7 cells. A limited in vivo study will greatly increase the impact of the findings.

2. It is not clear that why only one chemotherapeutic agent cisplatin was used.

3. The effects of deletion of six cGAS-STING pathway genes whose expression was found to be increased as well as other two, which were only increased for MCF7-J cybrids should be analyzed.

4.  The mechanisms by which cisplatin treatment leads to upregulation of ABCC1, BRCA1and CDKN1A/P21and downregulation of EGFR in MCF7-H and MCF7-J cybrids should be analyzed.  It is not clear why these effects remained unchanged in MCF7-H and MCF7-J cybrids.

5. Typographical errors were found throughout the manuscript and should be corrected by using professional editing service.

6. The authors should provide their own justification and relevance of the study. This will help the readers to understand the importance of the paper. 

Author Response

Reviewer#1(Comments to the Author):

Comment #1. Most of the experiments have been done in MCF7 cells. A limited in vivo study will greatly increase the impact of the findings.

Response: We appreciate the reviewer's valuable suggestion. The primary purpose of this study was to determine if cancer cell nuclei have a diminished ability to respond to mtDNA-altering signals via the cGAS-STING pathway.

Experiments are now being carried out to better understand the role of the cGAS-STING pathway in the cybrids of non-invasive versus aggressive breast cancer cell lines. Preparing the additional cybrid cell lines is a time-consuming and labour-intensive process. Our future plans are to perform knockdown studies on the various cGAS-STING genes using the relevant cybrid cell lines. Completion of these experiments will clarify the mechanism(s) of the cGAS-STING pathway in vitro. Moreover, we plan to conduct in vivo studies after we have gathered more positive results using other cybrids. However, at this point we believe that in vivo research on these mechanisms is beyond the scope of this paper, and we are working diligently on completing experiments focused on elucidating mechanisms for a second full-length manuscript.

Comment #2. It is not clear that why only one chemotherapeutic agent cisplatin was used.

Response: We thank the reviewer for bringing this to our attention. The current work was a continuation of our prior published research using the cybrid model [1,2] in which we assessed the effects of mtDNA variants representing individuals of European, African, Hispanic, and Asian maternal heritage in cybrid cell lines with the same ARPE-19 nuclei following treatment with cisplatin. Previous research has shown that an individual's mtDNA background may relate to variances in their response to cisplatin treatment, with retrograde signaling modulating the effectiveness and severity of adverse effects.

Cisplatin is used as first-line chemotherapy treatment for patients diagnosed with breast cancer along with other malignancies. Therefore, we solely used cisplatin response in our current study. The study's major goal was to see if retrograde signaling revealed a difference in response to cisplatin treatment in cancer and non-cancer cell nuclei. 

References: -

  1. Patel, T.H., et al., European mtDNA Variants Are Associated With Differential Responses to Cisplatin, an Anticancer Drug: Implications for Drug Resistance and Side Effects. Front Oncol, 2019. 9: p. 640.
  2. Abedi S, et al., Differential Effects of Cisplatin on Cybrid Cells with Varying Mitochondrial DNA Haplogroups. PeerJ. 2020 Oct 1;8:e9908.

Comment #3. The effects of deletion of six cGAS-STING pathway genes whose expression was found to be increased as well as other two, which were only increased for MCF7-J cybrids should be analyzed.

Response: We appreciate the reviewer's thoughtful comment. As we said in comment #1, experiments are currently being conducted to better understand the involvement of the cGAS-STING pathway in cybrids of non-invasive versus aggressive breast cancer cell lines. However, the procedure of creating these new cybrid cell lines is time-consuming and labour-intensive.

We intend to conduct knockdown studies on the multiple genes of the cGAS-STING pathway in different cybrid cell lines to better understand the role of cGAS-STING in relation to retrograde signaling in cancer cells. However, we believe that additional investigations into these mechanisms is beyond the scope of this publication and that the results presented within our manuscript can stand alone as a valuable contribution to the literature.  Experiments addressing the mechanisms are underway with the goal of a second full-length manuscript to describe the new studies.

Comment #4. The mechanisms by which cisplatin treatment leads to upregulation of ABCC1, BRCA1 and CDKN1A/P21 and downregulation of EGFR in MCF7-H and MCF7-J cybrids should be analyzed. It is not clear why these effects remained unchanged in MCF7-H and MCF7-J cybrids.

Response: We value the reviewer bringing this to our attention. Based on our previous findings, which showed that mtDNA copy numbers were similar in H and J cybrids, and therefore the differences in ATP generation and ROS formation were not attributable to lesser quantities of mitochondria within the cells but were more likely features of different mtDNA haplogroups modulating the nucleus (Kenney et al., 2013). Our previous published study (Patel et al., 2019) found that after cisplatin treatment, cells with J haplogroup (Northern European) variations exhibit distinctly different biological activity and gene expression patterns than cells with Southern European H haplogroup mtDNA.

While there was no difference in the quantities of ATP, lactate and mitochondrial membrane potential between the MCF7-H and MCF7-J cybrids in the current investigation, there were differences between the ARPE19-H and ARPE19-J cybrids (Patel et al., 2019). This may be one of the reasons for their similar gene expression patterns in MCF7-H and MCF7-J cybrids in response to cisplatin treatment. Further studies are needed to elucidate the mechanisms by which MCF7 cybrid responses are different from ARPE19 cybrids responses.

References:

  1. Kenney MC, Chwa M, Atilano SR, Pavlis JM, Falatoonzadeh P, Ramirez C, Malik D, Hsu T, Woo G, Soe K, Nesburn AB. Mitochondrial DNA Variants Mediate Energy Production and Expression Levels for CFH, C3 and EFEMP1 Genes: Implications for Age-related Macular Degeneration. PloS ONE. 2013 Jan 24;8(1):e54339.
  2. Patel, T.H., et al., European mtDNA Variants Are Associated With Differential Responses to Cisplatin, an Anticancer Drug: Implications for Drug Resistance and Side Effects. Front Oncol, 2019. 9: p. 640.

Comment #5. Typographical errors were found throughout the manuscript and should be corrected by using professional editing service.

Response: We extensively evaluated the manuscript and corrected all grammatical errors. If the reviewer finds additional grammatical errors, we will gladly correct them.

Comment #6. The authors should provide their own justification and relevance of the study. This will help the readers to understand the importance of the paper.

Response: We thank the reviewer once more for his excellent remark. This point has now been underlined in the revised manuscript's Introduction section. (Page# 2; Lines # 89-94)

Reviewer 2 Report

In this manuscript, the authors intended to investigate the role of retrograde signaling (mitochondria to nucleus) in MCF7 breast cancer cells.  The results showed that the MCF7-H and MCF7-J cybrids had identical metabolic/bioenergetic profiles and cisplatin responses. IThe retrograde signaling pathway in MCF7-H and MCF-J seemed to be altered.

1.    Please add the methods of APRE-related experiments, and the full name should be shown for its first appearance in the manuscript.

2.    In contrast with the activation of retrograde signaling patterns in cisplatin-treated APRE19 (J cybrids), the results shown in this study clearly indicated that the nuclei of cancer cells (MCF-7) were resistant to cGAS-STING signaling after cisplatin treatment. Although MCF-7 cybrids failed to activate cGAS-STING signaling, it could still decrease cell viability and ROS after cisplatin treatment. Please provide some discussion on the possible pathway in MCF-7 cybrids.

3.    Did the author screen the related gene in cisplatin treatment MCF-7 (not cybrids) to compare with MCF-7 cybrids?

4.  Writing (grammar) should be checked. For example, in Abstract:

Lines 26-27: In contrast, the MCF7-H and MCF7-J cybrids having identical metabolic/bioenergetic profiles and cisplatin responses. (no verb in the sentence)

Lines 27: “Our finding suggest that cancer cell nuclei may have diminished …..”

 Might be “Our finding suggests that cancer cell nuclei may diminish …..”

Author Response

Reviewer #2 (Comments to the Author): -

Comment #1. Please add the methods of APRE-related experiments, and the full name should be shown for its first appearance in the manuscript.

Response: We appreciate the reviewer's suggestion. We have included the ARPE-related experiments in the revised manuscript's Material and Methods section, as suggested by the reviewer. (Page #12, Lines # 411, 423; Page #13, Lines #472)

Comment #2. In contrast with the activation of retrograde signaling patterns in cisplatin-treated APRE19 (J cybrids), the results shown in this study clearly indicated that the nuclei of cancer cells (MCF-7) were resistant to cGAS-STING signaling after cisplatin treatment. Although MCF-7 cybrids failed to activate cGAS-STING signaling, it could still decrease cell viability and ROS after cisplatin treatment. Please provide some discussion on the possible pathway in MCF-7 cybrids.

Response: Once again, we thank the reviewer for the insightful comment. We have now highlighted this point in the Discussion and also provided more in-depth comparison of MCF7 cybrids to ARPE19 cybrids (Page 9, Lines 280-292 and Page 9-10, Lines 293-315).

Comment #3. Did the author screen the related gene in cisplatin treatment MCF-7 (not cybrids) to compare with MCF-7 cybrids?

Response: We greatly appreciate the reviewer's recommendation. The primary goal of this study was to compare the responsiveness of retrograde signaling in non-malignant (ARPE19) versus cancerous (MCF7) cybrids to the effect of cisplatin treatment. As a result, we did not screen the expression of associated gene expression of MCF7 wild-type cells in response to cisplatin treatment and compare it to MCF7 cybrids in the current study. However, we appreciated the reviewer's idea and will incorporate it into our next full-length publication, in which we will examine the differential response of different cybrids produced from primary and metastatic breast cell lines versus the corresponding parental cell line.

Comment #4. Writing (grammar) should be checked. For example, in Abstract: Lines 26-27: In contrast, the MCF7-H and MCF7-J cybrids having identical metabolic/bioenergetic profiles and cisplatin responses. (no verb in the sentence).  Lines 27: “Our finding suggest that cancer cell nuclei may have diminished.” Might be “Our finding suggests that cancer cell nuclei may diminish.”

Response: We extensively evaluated the manuscript and corrected all grammatical errors. If the reviewer finds additional grammatical errors, we will gladly correct them.

Round 2

Reviewer 1 Report

The authors have addressed al my concerns.